# Move-It: A Cluster-Randomised Digital Worksite Exercise Intervention in China: Outcome and Process Evaluation

**DOI:** 10.3390/ijerph16183451

**Published:** 2019-09-17

**Authors:** Holly Blake, Betsy Lai, Emil Coman, Jonathan Houdmont, Amanda Griffiths

**Affiliations:** 1School of Health Sciences, University of Nottingham, Nottingham NG7 2HA, UK; 2Nottingham NIHR Biomedical Research Centre, Nottingham NG7 2UH, UK; 3School of Medicine, University of Nottingham, Nottingham NG7 2UH, UK; betsy.hk@gmail.com (B.L.); jonathan.houdmont@nottingham.ac.uk (J.H.); amanda.griffiths@nottingham.ac.uk (A.G.); 4Health Disparities Institute, University of Connecticut, Hartford, CT 06030-7030, USA; coman@uchc.edu

**Keywords:** worksite, exercise, physical activity, health promotion, sedentary, digital

## Abstract

We evaluate the outcomes and processes of a video and web-based worksite exercise intervention for sedentary office workers in China, in a 2-arm cluster-randomised wait-list control trial (*n* = 282: intervention (INT) *n* = 196 and wait-list control (WLC) *n* = 86). Eligible clusters were two sites of a major organisation in China randomly allocated to each group (INT: Guangzhou; WLC: Beijing); eligible participants were site employees (*n* = 690). A theoretically informed digital workplace intervention (Move-It) involving a 10 min Qigong exercise session (video demonstration via website) was delivered twice a day at set break times during the working day for 12 consecutive weeks. Individual-level outcomes were assessed. Participants’ physical activity increased significantly from baseline to post-intervention similarly in both the intervention and the control group. There was a significantly smaller increase in weekday sitting hours in intervention than controls (by 4.66 h/week), and work performance increased only in the control group. Process evaluation (including six focus groups) was conducted using the RE-AIM (reach, effectiveness, adoption, implementation and maintenance) framework. The intervention had wide reach and was successfully marketed to all employees with good uptake. The participatory approach increased perceived organisational support and enhanced adoption. The intervention was implemented broadly as planned. Qigong worksite exercise intervention can be successfully delivered to sedentary office workers in China using video and web-based platforms. It may increase physical activity and does not adversely affect perceived work performance. The study highlights the complexity of conducting health promotion research in real-world organisational settings.

## 1. Introduction

China is the most populous country in the world with a current population of approximately 1.42 billion. Promoting population health has become a national priority [1,2]. The United Nation’s Sustainable Development Goals for 2030 include reducing premature mortality from non-communicable diseases by one third, through targeting risk factors for chronic disease. Analysis of data from the Global Burden of Disease Study 2013 has been used to make projections for China in line with this goal [3].

In recent years, there has been a marked increase in the number of overweight/obese adults in China, particularly those living in urban areas [4]. Physical inactivity is prevalent in China [5,6] and is associated with population increases in chronic disease risk factors such as hypertension and diabetes; in 2017 more than 245 million people in China were estimated to have hypertension [7] and 114 million, diabetes [8]. Sedentary behaviour (SB) is associated with adverse health outcomes independently of levels of physical activity (PA) [9]. Occupational sitting is a health hazard for office workers [10]. Prolonged sitting of >60 min without interruption is associated with increased risk of metabolic syndrome, obesity and cardiovascular disease [11]. In China, the overall prevalence of high SB was 8.3% in a cross-sectional study using nationally representative data from six low- and middle-income countries, including China [12]. A study in three major metropolitan cities in China showed that over 80% of the participants were sedentary (participating less than 90 min/week in leisure PA) [13]. The strongest predictor of the rapidly increasing prevalence of overweight in China is engagement in light rather than heavy work-related PA [14]. In Chinese urban populations, a high proportion of Chinese adults engage in no leisure-time PA at all [15] (67% of females; 61% of males) and consequently, occupational activity is the major source of PA.

In 2018, sixty-six per cent of the population in China was in employment [16] and therefore workplaces offer a platform for broad reach of health promotion efforts in line with national public health policies. Additionally, China has faced labour shortages in both skilled blue-collar and white-collar jobs due in part to economic growth in domestic and export markets [17]. Attracting and retaining workers, also known as the “war for talent”, has become a high priority for human resources management. Promoting employee health through corporate initiatives may also represent an important component in organisational efforts to improve labour retention [18]. Albeit of moderate quality, there is some evidence from a recent systematic review that workplace interventions (including employee exercise interventions) may improve factors of importance to employers [19].

The vast majority of evidence for workplace health interventions has been gathered from Western settings. Commissaris and colleagues [20] conducted a review of the effectiveness of workplace interventions that are implemented during productive work and are intended to change workers’ SB and/or PA. The review included 40 studies describing 41 interventions organised into three categories: alternative workstations (*n* = 20), interventions promoting stair use (*n* = 11), and personalised behavioural interventions (*n* = 10). The review showed positive effects for PA or SB—alternative workstations decreased overall SB (strong evidence, even for treadmills separately); interventions promoting stair use increased PA at work while personalised behavioural interventions increased overall PA (both with moderate evidence). Evidence was either insufficient or conflicting for intervention effects on work performance and lipid and metabolic profiles [20]. Another recent review highlights potential benefits of worksite exercise training for workers (decreasing health risk indicators, improving physical capacity and functions as well as perceived health) and employers (decreased sickness absenteeism and presenteeism in terms of improved or maintained productivity and work ability) [21].

Benefits for health and employee productivity have been reported in Chinese settings from sit-stand workstations [22], pedometer walking challenges with prompts [23,24] and multicomponent health promotion programmes [25] (although performance was not measured in these studies).

Previous studies have demonstrated the feasibility of integrating exercise breaks with the work schedule as a strategy to promote worker health [26]. Short exercise breaks in the workplace, incorporated within daily organisational routines, may provide an economical and practically viable option [27], although organisations often have real-world concerns about the approach and investment required. Technology-based interventions are one approach to deliver health promotion with a potentially broad reach and low cost. There have been advances in the use of digital technologies designed to intervene with office workers’ health behaviours (e.g., SB [28]). There is modest evidence that digital-only interventions have a positive impact on health-related outcomes in the workplace; a review of 22 randomised controlled trials showed that significant improvements were found for a broad range of health outcomes including SB and PA levels [29]. In the Chinese population there is low uptake of the use of technology to support health and healthcare more generally [30]. Simple technologies such as video materials and health websites remain novel for health promotion initiatives, and overall, workplace interventions designed to promote healthy lifestyle behaviours remain far less prevalent than in Western settings.

Therefore, there is a need for research to establish whether technology-based interventions delivered in workplaces in China can improve employees’ individual overall PA, and influence work-related outcomes. The focus is on using the workplace as a mechanism for reaching large numbers of working-age adults for delivery of PA intervention, with the intention of getting workers standing up and more active during the working day, and improving their overall PA (including both work-related PA and leisure-time PA).

Qigong is practiced extensively in China and is therefore a culturally appropriate and widely known activity. Practicing Qigong in groups is commonly undertaken to help promote team building and social support. There is a growing body of evidence for the therapeutic effects of Qigong, with regard to general health benefits, body composition, chronic disease prevention and management [31,32,33]. It is a system of gentle physical exercises and breathing control related to Tai Chi, which could be delivered in a worksite setting, at low cost, individually or with peers. This mindful approach to exercise promotes mental concentration and positive mental wellbeing, and this is important since employee mental wellbeing influences business performance [34]. In light of this mindful approach, we chose to focus on potential impacts on work performance as a measure of operational efficiency (e.g., the way in which someone functions to accomplish something successfully), rather than productivity per se (e.g., indicators of output).

Studies have reported on the increase in engagement in Qigong in U.S. office workers [35], but to our knowledge, there are no published studies evaluating the effectiveness of worksite Qigong intervention in China. Although digital platforms have been used in the Chinese workplace setting to promote other areas of health (e.g., sleep hygiene [36]), there is a paucity of technology-based approaches to promote workplace PA in China, and very few studies have reported on work-related outcomes of importance to organisations, such as worker performance [37]. Importantly, more knowledge is required on the implementation of technology-based PA interventions in the workplace setting in China.

### Aim

The aim of the study was to deliver and evaluate a digital video-based worksite exercise intervention in China. The objectives were (1) to examine the effect of a video-based intervention on PA and weekday sitting hours (individual level); (2) to examine the effect of a video-based intervention on individual work performance (individual level); and (3) to conduct a process evaluation of the intervention implementation (individual and cluster level). We hypothesised that there would be an increase in PA, a decrease in sitting time and an increase in work performance following exposure to the intervention.

The RE-AIM (reach, effectiveness, adoption, implementation and maintenance) framework [38] was used to guide outcome and process evaluation, assessing intervention reach, effectiveness, adoption, implementation and maintenance. Reporting across RE-AIM dimensions has been identified as an important first step to enable the effective translation of interventions into real world settings [39].

## 2. Methods

### 2.1. Design

This was a 2-arm cluster-randomised wait-list control trial, with a 12-week intervention and outcomes assessed in two groups: intervention (Guangzhou), and wait-list control (Beijing) (Objectives 1 and 2). Wait-list design was selected to ensure an ethical approach for the intervention to be accessible to all company employees; this was also a requirement from the participating organisation, since it reflects a pragmatic real-world workplace health promotion approach. Cluster-randomisation (rather than individual randomisation) was required due to the nature of the intervention: there was a worksite participatory approach to intervention delivery, an option for group-based engagement with the intervention, and co-workers were involved in provision of support for PA. Reporting was in accordance with the CONSORT Extension for Cluster Randomised Trials (see Appendix A), the Template for Intervention Description and Replication (TIDieR) checklist (see Appendix A) and the Intervention Fidelity Plan for Qigong Interventions [40]. The trial is registered on clinicaltrials.gov (NCT04038658).

### 2.2. Methods

#### 2.2.1. Outcome Evaluation: Participants and Procedures

Eligible clusters were two sites of a large Information Technology (IT) organisation in the private sector, located in two major cities in China (Beijing and Guangzhou). Eligible participants were employees of the organisation. The research team was independent from the participating organisation. The IT organisation was not involved in the study conception or design and had not financially supported the research or research team. Given the gentle nature of the form of Qigong adopted, we took a complete enumeration approach in that all employees were invited to take part, which reflects practice in real-world workplace health promotion programmes. Employees could choose to take part, or to decline participation if physical exercise was contraindicated (e.g., pregnancy or co-morbidity precluding exercise). Information about the nature of the exercise was provided in the intervention materials, and there were additional opportunities for employees to discuss possible contraindications with trained team leaders or the Qigong Master. The sites were similar in terms of employee age, gender ratio and job role profiles. Employees at both sites were predominantly local Chinese, aged 25–40 years, and the majority were university graduates. The majority (75%) of participants were computer programmers and project managers, frequently involved in conference calls with clients. Contracted hours were a central defining characteristic of all employee job roles within the participating organisation.

The project researcher obtained cluster-level informed consent prior to randomisation from relevant Human Resource (HR) officers. An organisational committee (two team leaders and two HR officers) was set up to develop and implement company policy on the internal marketing of the intervention. To demonstrate senior management commitment, the general manager and HR manager delivered a standard 30 min orientation and motivation briefing session to all team leaders before the intervention began.

The HR department invited all employees from both sites to participate in the study via their company intranet webpage. This provided all employees with a link to information about the study; individual participants indicated their informed consent to participate electronically by clicking on a yes/no box. Employees were informed that taking part was voluntary and that they could decline or withdraw from the research at any time. The study was conducted in accordance with the Declaration of Helsinki. The protocol was approved by the Ethics Committee of the Institute of Work, Health and Organisations, University of Nottingham, UK on October 31st, 2012 (Lai/27613). Baseline data for all consenting employees (hereby referred to as participants) were collected by the study researcher prior to site allocation to intervention or control group, and so group allocation was not known to clusters or individuals at this point. Data were collected through online questionnaires emailed to all participants in March 2013 (Time 1: T1). Participants in both groups completed a second online questionnaire in July 2013 (Time 2: T2). Time allocated for responses was two weeks. For the intervention group, T2 (October 2013) was immediately post-intervention, and for the wait-list control group this was just prior to receiving the intervention. This allowed for comparison of the intervention group with an untreated comparison, while at the same time allowing wait-listed participants to benefit from the intervention.

The two sites were then randomly allocated to intervention/control groups by a senior manager who was not part of the research team, at a 1:1 ratio using a computer-generated random number sequence, assigning one site as the intervention group (*n* = 490 employees) and the other as the wait-list control group (*n* = 200 employees). Regarding allocation concealment, both the research team and participants were therefore blind to group allocation prior to baseline data collection. Individual randomisation was not possible due to the risk of group contamination. Since the intervention was implemented across the whole workplace it was not possible to withhold the marketing of this initiative (and therefore intervention exposure) from individual employees. Based on an assumption of seven extra hours in International Physical Activity Questionnaire (IPAQ) changes, i.e., an increase of five hours in controls and 12 h in intervention participants across the study period (with a SD = 15), the estimated sample size was a minimum of 72 for the intervention group and 72 for the control group, for alpha 0.05, beta 0.2 and power 0.8 (done at https://clincalc.com/stats/samplesize.aspx, see Appendix A). An increase in 12 h in the intervention group could be achieved by adherence to 60% of the intervention “dose”. Due to the nature of the intervention, it was not possible to blind sites or participants to group allocation. The study flow is shown in Figure 1. In the end, 282 employees participated, 196 in the intervention group and 86 in the control group.

#### 2.2.2. Process Evaluation: Participants and Procedures

All participants in the process evaluation were recruited from the intervention site. Six focus groups, facilitated by the researcher (BL), took place on completion of the intervention, between October 2013 and January 2014. These involved employees (both participants and non-participants), organisational committee members and senior management. Written informed consent was obtained from all participants. All focus groups were conducted in Chinese. Focus groups with employees and the organisational committee were audio recorded with permission, transcribed verbatim, anonymised and translated into English by a bilingual researcher. An independent translator, fluent both in English and Chinese, helped to validate the transcripts by repeated listening to all focus group recordings. In the event of any disagreement about the translation, the researcher met with the translator to discuss and agree changes to the final version of the transcripts.

In focus groups with senior management, audio recording was not possible, and therefore detailed notes were taken by the researcher. Observations of non-verbal behaviours and group interactions were noted in all sessions. Informal feedback offered by managers not attending the sessions was also recorded. Minutes were taken in project meetings, and all intervention promotional materials collated.

##### Focus Groups with Employees

Four focus groups were held with employees: three with intervention participants (*n* = 7–9) and one with non-participants (*n* = 5). Employees from several departments were invited to participate by an HR manager, on the basis of their availability at the scheduled time, and thus constituted convenience samples. They were briefed by the researcher and written consent obtained. Focus groups took place in a meeting room on site and lasted on average 45 min. A semi-structured interview schedule was used to stimulate discussion about factors impacting on employee participation or non-participation, informed by the literature on worksite PA interventions [41]. Questions probed the reasons for participation or non-participation, and for those who participated, whether the programme met their expectations, the perceived impact of the programme and the potential advantages and disadvantages of continuing the programme as a long-term workplace policy.

##### Focus Group with the Organisational Committee

A focus group with three of the four members of the organisational committee was conducted. The focus group took place in a meeting room on site and lasted for 60 min. Questions in a semi-structured interview aimed to capture committee members’ perceptions of the intervention and experiences of the implementation process. Questions probed views about barriers and facilitators to delivering the intervention, visible demonstrations of company commitment and possible reasons why participants adhered to (or failed to adhere to) the exercise programme. They were also invited to provide suggestions as to how the initiative might be maintained in the long-term.

##### Focus Group with Senior Management

A focus group was conducted with two members of senior management, together with the general manager and the HR manager. The focus group took place in a venue near to the organisation and lasted for 60 min. Semi-structured interview questions aimed to explore motivations for sponsoring the initiative and promoting workplace PA [42] and views on demonstrations of company commitment. The group was also invited to provide suggestions as to how the initiative might be maintained in the long-term.

### 2.3. The Move-It Intervention

In line with the Intervention Fidelity Plan for Qigong Interventions [40], we report on the intervention design, training of instructors, intervention delivery and receipt, as well as enactment. This digital intervention consisted of a series of six video clips demonstrating Qigong exercises designed to be undertaken twice per day (for 10 min), on every working day, at set exercise break times. Qigong is an ancient Chinese exercise that involves deep abdominal breathing and stretching. The exercise series was designed by a Chinese Qigong Master with over 20 years’ experience. A unique feature of this study was the co-creation of video materials with participating sites; the Qigong session protocol was developed by the Qigong Master, and team leaders (*n* = 4) were involved in the development of the video clips by modelling the exercises themselves, under direct supervision of the Qigong Master. This approach was undertaken at both sites, such that employees could see the exercises being modelled by familiar faces from their own management team (culturally, a broadly homogenous group). Each site produced a set of six videos, each video lasting for approximately two minutes with a total of 10 min of exercise content. Therefore, the total “per-protocol” dose was 1200 min (20 h). The six videos each represented one module: neck, shoulder, arms, waist, legs and whole body. Each module had three Qigong movements (demonstrated in the two-minute video), and each movement was repeated by participants three times (resulting in 10 min exercise content). The Qigong protocol accounted for differences between the Qigong Master and team leaders in terms of education, experience and learning style. For participants, practice (duration and frequency and intensity (low to moderate) of the movements) could be tailored to suit individual needs (e.g., younger or older, experienced and less experienced, or those recovering from illness or injury). Such short exercise interventions have previously been employed successfully in interventions designed to encourage behavioural change [27].

The company’s organisational committee branded the intervention as ‘Move-It’. To align with a clear branding and communication strategy, known to facilitate workplace PA campaigns (e.g., [43]), a specially designed logo and selected images were used on promotional posters and a dedicated Move-It webpage was established, co-developed by the researchers and the organisational committee (cluster level activity). The six exercise videos were designed to be posted in sequence, one every two weeks, on the Move-It website. The intention was that employees would follow a sequential pattern of exercise, mastering one exercise done repeatedly across two weeks to complete a full body programme amounting to the 20 h of exercise content. The intervention was designed to be completed within 12 weeks; this is because habit formation (based on daily repetition) is expected to take approximately 10 weeks, with an estimated 2–3 months for a behaviour to become “second nature” [44].

The intervention was developed using the Behaviour Change Wheel (BCW) approach [45], to understand the behaviour, identify intervention options, identify content and then implementation options. We sought to understand the staff’s capability, opportunity and motivation to engage in the behaviour using COM-B (Capability–Opportunity–Motivation-Behaviour) [45]. The BCW links COM-B components to nine intervention functions through which an intervention can change behaviour, and seven broad policy categories relating to decisions authorities can make to facilitate delivery of intervention functions. We used the APEASE criteria (affordability, practicability, effectiveness, acceptability, side effects and equity) [45] to select the most appropriate intervention functions to change. The six COM-B components and the four selected policy categories are mapped to the six selected intervention functions in Table 1.

Regarding the COM-B components, physical capability (C-Ph) is achieved through Qigong physical skill development via employee training and modelling of Qigong movements, and train-the-trainer sessions for demonstrators. Psychological capability (C-Ps) is achieved through imparting knowledge via web and video materials, building self-efficacy for exercise, and building physical activity skills through training. Reflective motivation (M-Re) is achieved through increasing employee understanding about the benefits of physical activity for health and wellbeing, eliciting positive feelings about workplace Qigong exercise. Automatic motivation (M-Au) is achieved through associative learning that elicits positive feelings and impulses and counter-impulses relating to physical activity behaviour, exercise habit formation and direct influences on automatic motivational processes (e.g., via on-screen prompts). Provision of Qigong demonstration videos allows direct imitation of movements. Physical and social opportunity (O-Ph; O-Soc) is achieved through organisational changes to create sanctioned and scheduled exercise breaks (reducing barriers to exercise), and provision of social support via group-based exercise.

Regarding policy, organisational changes include communication and marketing strategies using print and online media; guidelines in the form of the Qigong exercise protocol; regulation in the form of established rules around exercise practice at set times in the working day; and service provision through IT support for on-screen prompts, marketing, communications and website educational materials.

The APEASE criteria [45] were used to select appropriate behaviour change techniques (BCTs) that were relevant to the six identified functions, using definitions included in the BCTTv1 [46]. These BCTs were then translated into intervention content. The Template for Intervention Description and Replication (TIDieR) checklist [47] was used to specify details of the intervention including the who, what, how and where of proposed intervention delivery.

### 2.4. Intervention Delivery

The intervention was delivered at both the cluster and individual level. A participatory approach to intervention delivery was established. Team leaders (*n* = 31) acted as intervention facilitators; an organisational committee of senior managers and HR representatives were responsible for overseeing internal marketing as well as delivery of orientation and motivation briefing sessions for the team leaders; and employees from the staff-side contributed their assistance with technical support, production and review of materials and exercise demonstrations. Team leaders were invited to volunteer by the organisational committee. The role of the team leaders was to facilitate intervention delivery and encourage co-workers to engage with the intervention. They were asked to model the sequence of movements during each set exercise break session, with random sessions directly observed by the Qigong Master. Team leaders were already familiar with Qigong approaches, although prior to the study start, a one-hour “train-the-trainer” session was conducted, by the Qigong Master, with the team leaders at both sites of the organisation. The purpose was to provide a standardised approach for intervention delivery at cluster level, whereby managers were provided with a clear demonstration of the exercise movements they were required to model, and they received the same communication around operationalising (and supervising) the intervention with employees (assessed by the Qigong Master). The duration and content of the training session had been established prior to study start through a process of stakeholder consultation including members of the participating organisations, the Qigong Master and the project team.

Educational materials, the Qigong movement video, and instructors’ feedback were used to facilitate participants’ comprehension. The researcher provided team leaders with a paper copy of the exercise instructions (treatment manual) and the exercise demonstration videos. To further ensure understanding and comprehension of the Qigong protocol (a) team leaders were able to contact the Qigong master to discuss intervention sessions, or any observed problems to maintain a high standard of delivery and prevent intervention drift, and (b) participants were able to contact team leaders to discuss Qigong challenges or barriers to practice at work, or at home. The videos were also uploaded onto the company’s Move-It website and were available to all employees as the standard reference (cluster level activity). A Move-It icon on each employee’s desktop computer screen was scheduled to pop up twice a day, at the same time every day: 10:50 and 15:50. This acted as a sign-in and a prompt for employees to interrupt prolonged sitting and to individually perform the exercise routines (individual level activity) beside their usual workstation. In acknowledgement of the characteristics of the work undertaken by this group of workers [48], they could choose to participate in an exercise group led by their team leader at the allocated times, or to ignore the prompt and participate individually at their workstations, guided by a video, at a time of their preference.

### 2.5. Measures

Self-report measures of PA, work performance and weekday sitting hours were used at T1 and T2 (individual level outcomes). Questionnaires were provided in both Chinese and English since it was the company practice to have all written communication available in both languages. Public involvement consultations at protocol development stage indicated that employees and managers preferred single-item or short-form measures in the online questionnaire, to maximise completion and reduce disruption to daily work patterns.

#### 2.5.1. Physical Activity

The IPAQ (International Physical Activity Questionnaire), Taiwanese short form [49] was used to assess self-reported overall PA. Participants were asked to estimate the number of days they performed vigorous, moderate and walking activities (frequency) and the time (duration) spent doing these activities in in the previous seven days. The total PA for each participant was calculated as the sum of walking, moderate and vigorous PA, expressed in terms of metabolic equivalent hours per week (MET-hours·wk^−1^). Higher scores indicated higher levels of PA.

#### 2.5.2. Work Performance

This was assessed by a single item from the World Health Organization Health and Work Performance Questionnaire (HPQ) [50]. This item was measured on a 10-point scale ranging from 0 = worst performance to 10 = top performance (i.e., “how would you rate your overall performance on the days you worked during the past 4 weeks?”). Higher scores indicate greater perceived work performance.

#### 2.5.3. Weekday Sitting Hours

Participants were asked to indicate the number of hours (to the nearest whole hour) spent sitting on a typical workday (including sitting at home and work, and during travel). Higher scores indicate a greater number of daily hours spent sitting.

#### 2.5.4. Adherence and Adverse Events

Information on exercise adherence was collected by the participating organisation using an interactive computer-based system. Individual daily exercise logs (frequency) were recorded automatically once participants clicked on their Move-It screen icon after being prompted to participate. These individual-level exercise adherence data were collected by the participating organisation, although only cluster-level summary reports were accessible to the research team.

Team leaders made field notes, recorded any Qigong-related adverse events and reported them to the research team, defined in accordance with the literature [51]. They were required to report adverse events immediately to the research team. Adverse events were defined as “a variety of undesirable experience or any slightly unfavourable and unintended sign, feeling, symptom, physical and mental changes or disease that participants endure during or after treatment or intervention with Qigong training regardless of causal relationship, but are not serious to the point of affecting normal life and work”. Serious adverse events (resulting in stopping the trial) were defined as “an event leading to serious outcomes such as being life-threatening, permanent damage, require either in-patient hospitalisation or the prolongation of hospitalisation, results in persistent or significant disability/incapacity or death” [51].

### 2.6. Data Analysis

Intervention outcome data were analysed using Stata 16 (StataCorp, TX, USA) [52] by an independent statistician who had not been involved in intervention delivery or data collection and was blind to group allocation. We tested the normality of the pre and post outcomes, and all three outcomes were skewed and kurtotic (except baseline job performance), with an excess of higher values; however, instead of transforming them, we tested whether analyses specifically modelling the non-normality would yield different results than those assuming normality, using the Mplus skewt recently implemented modelling option (see an example with software code posted in Anderson and colleagues [53]). The skewt option of estimation simply adds two parameters (besides mean and variance) for all variables analysed, and hence properly corrects estimates for non-normality. In order to ascertain whether improvements had occurred between the pre- and post-intervention time points we used 2-group latent change score (LCS) models, and intuitive structural equation models (SEM) which have been shown to be extensions of the paired samples t-tests [54]. Statistical significance was defined as *p* < 0.05. Data from both intervention groups were included in all analyses, and analysis was from an intention-to-treat perspective, by means of maximum likelihood estimation of changes for the entire sample, including those who did not fill out post measures. The full information maximum likelihood algorithm (FIML) estimates parameters for the entire sample even when some cases have missing values on one of the variables, by using all data points when maximizing the likelihood of observing the data, given the parameters. Note that SEM with FIML can “back fill” values for the missing post outcome, by estimating “factor scores”, i.e., latent variable individual scores for the latent change score, such that then by mere differencing the post values can be attached to the raw data (post = change − pre). Stata and Mplus code and outputs are available in Appendix A and posted online at http://bit.ly/exercise.blake.

A deductive approach informed the analysis of transcripts and meeting notes, using the five dimensions of the RE-AIM [38] framework: reach, effectiveness, adoption, implementation and maintenance. Within these broad categories, thematic analysis was used to identify patterns in the transcripts and meeting notes [55].

## 3. Results

### 3.1. Reach

Reach is defined here as the absolute number, proportion and representativeness of individuals who were willing to take part in the study. A total of 490 employees at the intervention site were invited to participate, with 196 completing the baseline survey (40% response rate). Among the 200 employees at the control site that were invited to participate 86 completed the baseline survey (43% response rate). Baseline participant characteristics are displayed in Table 2. Attrition analyses (*n* = 282) showed that there were no significant differences between those who returned (*n* = 214) and those who did not return for the post-test (*n* = 68) in terms of gender, marital status, education or experience, nor when comparing total MET and sitting time. Returners, however, had higher average job performance at baseline (7.25) compared to those who dropped out (6.68), which might indicate that dropouts may have been less motivated overall; this outcome was tested therefore for difference in changes using an additional model in which baseline job was a predictor of the changes.

The sample was broadly representative of the overall workforce in terms of gender, age, marital status and length of time working for the company. There were no significant differences between the characteristics of participants and non-participants in terms of gender, age and work experience in the organisation. There were no baseline group differences in physical activity and job performance either, but the intervention group had lower average sitting hours per week (6.89 h/w) than controls (7.63 h/w, t(275) = 4.008, *p* < 0.001). However, participants in the control group were more likely to be university educated, and married, than participants in the intervention group (see Table 2). Loss to follow-up was 27% in the intervention group, and 15.1% in the control group. Analyses were conducted on data contributed by study participants that completed the T1 (pre-intervention) surveys, i.e., the full sample, resulting in an analytic intention-to-treat sample of 196 intervention group participants (of whom 73% returned for T2) and 86 in the control group (85% of whom completed the T2 surveys).

Focus group data showed that participants perceived the internal communications about the intervention to be very successful. The success of the programme marketing and interest from staff was evident from many accounts. The general view expressed was that information about the programme reached everyone in the organisation and that the formal participation rate (40%) was likely to be an underestimate. One participant commented:
“This programme definitely got 100% attention rate. Everyone from the top management to the junior staff knew about ‘Move-It’. And all levels of employees participated.”

The intervention appeared to filter through to employees who were not formally enrolled as participants, and therefore the reach of the intervention was perceived to be much wider than the number of employees that provided their electronic consent and completed the study questionnaires. An organisational committee member commented:
“Even though some employees did not enrol on the programme for whatever reasons, they watched the videos and practised the exercises together.”

### 3.2. Effectiveness

Table 3 displays the changes in each of the outcome variables between T1 and T2 for the intervention and control group, respectively, as estimates of the latent change score intercepts, i.e., the average changes controlling for where the participants started at baseline. These intercepts are the average changes for someone who starts at baseline at an average (control) outcome value (predictors have to be centred for intercepts to be meaningful).

The intervention did not result in greater changes in the intervention group than in controls. There was an increase in hours of PA per week in both the intervention group (5.80 h/w) and the control group (7.41 h/w), and the changes were statistically significant in both groups (a non-significant difference in the changes between groups, *p* = 0.70).

Work performance appears to have increased significantly more in the control group (0.69 units average increase) than in the intervention group (−0.03 units average drop).

Sitting hours increased in the control group by 10.34 h/w and in the intervention at a lower 5.68 h/w. The difference in changes (−4.66 h/w) was statistically significant (*p* values are in the last column in Table 3).

We note also that the extent to which where one starts impacts how much they change differed for the job performance and sitting outcomes, but not for physical activity (data available in Appendix A).

There were no adverse events reported by employees at either site. Objective data on exercise adherence were collected by the participating sites using computerised daily exercise logs. The participating organisation reported a decline in adherence during the intervention period (T1 to T2). However, we do not know the number of times each participant actually took part in the exercises. The raw adherence data were not made accessible to the research team and so we are unable to present data on adherence rates, although participating sites reported that the decline in adherence that they observed may have been influenced by (i) participants forgetting to “click” on the icon during the exercise routines, and (ii) where participants practised as a group, only one of them clicking on the icon.

Many participants perceived the Qigong exercises positively and reported positive benefits on physical and mental health including muscle relaxation, stress reduction and improved working mood:
“It helps with neck and muscle pain, which seems to be common among our colleagues.”;
“Exercise makes me feel good...more positive and energised.”

Some employees reported that this health initiative aligned with corporate values, raising awareness about workplace health and wellbeing, and serving as an indication that the organisation cared for the people. Attracting and retaining skilled workers appeared to be quite salient in the IT industry in China:
“It helped to build employees’ awareness of the importance of health at work.”;
“Staff would feel that we care for their well-being. It might help with stronger sense of belonging… staff retention perhaps.”

### 3.3. Adoption

Adoption is defined here as the absolute number, proportion and representativeness of settings and employers willing to deliver the intervention, the level of organisational support for delivery and employee engagement in the intervention.

In this wait-list design, the intervention was successfully delivered in both sites at different times, therefore the whole of the participating IT organisation (total clusters/sites = 2; total employees = 690; organisational committee = 4; management = 4; and team leaders = 31). Team leaders in the intervention and control group all attended the 30 min orientation and motivation briefing session delivered by the management team (100% attendance).

In the focus groups, the majority of participants spoke positively about senior management and it was perceived that high organisational support was provided for the duration of the project, including during the set-up period for the intervention. For example, management were involved in the design and development of promotional posters and exercise videos. Promotional materials were considered by all of the focus group participants to be of high quality. The Move-It posters were placed in high visibility areas, including office corridors and the staff canteen; the promotional videos were shown on a large screen in the canteen. Participants also commented that management provided good technical support throughout the study period, including setting up the development of the pop-up window system and online exercise log recording system.

Some barriers to adoption were noted. Although team leaders were designated by the organisational committee to model the sequence of movements during each exercise break session, not all leaders were thought to have adopted this role fully. Focus group participants suggested that it might have been better to appoint a designated person committed to lead the sessions, whether they were a team leader or not.

It was reported that there was difficulty in exercising in groups for some teams because of limited space in their office environment. However, in general, it was reported that participants preferred to practise the exercise routines together in groups rather than on their own. Peer support was therefore identified as an important facilitator of intervention adoption, and conversely, a lack of peer support as a barrier to engagement. One participant commented:
“The mood was contagious. It was boring to do the exercises alone. For myself, when I saw the pop-up window and nobody nearby participated in the exercises, I didn’t do them.”

Many described that the intervention helped raise employees’ awareness of the importance of health at work. The participatory approach was thought to be key to its success. Employees reported enjoying the camaraderie developed during the design and implementation process. An organisational committee member commented:
“Many enthusiastic employees participated in the project, including in the production of the promotional videos and with technical support to place the videos onto the company intranet.”

The limited data collected by the participating company using online exercise adherence logs had demonstrated declining adherence rates in the intervention group over time, although this did not necessarily concur with the views of focus group participants about individual employee behaviours. According to participants, adherence to the exercise intervention was exceptionally high; for some individuals adherence remained high for the duration, for others, adherence was more noticeable during the first few weeks.

Perceptions of the study materials were positive and thought to encourage intervention take-up. Comments from participants indicated that the videos were seen to be attractive, easily accessible, user-friendly and enjoyable. There was a general consensus that the fact that the videos were tailor-made for the study sites was helpful. Some found the video demonstrations amusing since they contained demonstrations from familiar managers and colleagues.

The most common reason why employees did not formally enrol in the study was reported to be fear of committing themselves to too many activities that might impact on their work-life balance. Another view was that there were already too many surveys for employees to complete during the working day.

Some participants found the interruption from the pop-up screen disturbing when they were particularly busy and concentrating on their work. The flexibility of being able to ignore prompts if required was considered useful. Nonetheless, periodic prompts were still considered necessary (and important) to interrupt people from prolonged periods of sitting. An organisational committee member said:
“They needed to be prompted, whether by machine or in person. They needed to be prompted to stop their current work and take an exercise break.”

### 3.4. Implementation

Implementation is defined here as intervention fidelity; that is, whether the delivery of intervention adhered to the plans. Regular assessment of intervention fidelity is recommended to ensure delivery is as fully compliant with the original implementation plan as possible [56] and that any deviations are recorded. Since there was only one site in the intervention group there was no between-site (within-cluster) inconsistency in delivery (e.g., in costs, adaptions or delivery processes).

Focus group participants considered that the intervention was delivered broadly as planned. One exception concerned the fact that all the standard exercise demonstration videos were posted to the intervention website at the start of the intervention period, rather than at two-weekly intervals as originally planned, alongside the managers’ promotional videos. Therefore, although the employees received all of the intervention content, we cannot guarantee whether participants chose to access them in the same order or focused on mastering one exercise before viewing the next. A participant observed:
“The majority of participants practised the full version of exercise movements in the first two weeks.”

Further, it was perceived that there was potential for contamination between intervention and wait-list control site groups. This was because the intervention preparatory work started much earlier for the wait-list control group than it had for the intervention group; for example, in team leaders’ involvement in the production of promotional videos (from both sites) which happened prior to intervention delivery at the first site.

It was noted that sometimes the exercise sessions lasted longer than the designed time frame as participants chose to repeat each movement several times. A few participants reported that because of work commitments, they practised the exercise at their own convenience rather than at the scheduled times. Some mentioned practising independently at home with their children as well as in the workplace. Some participants requested online access to the exercise demonstration at home so they could practise with their families. Although some of the participants were therefore not following the prescribed exercise scheduling, it does demonstrate that the online adherence data collection (albeit with our access to trends only) may have underestimated employee engagement in the intervention. Indeed, the engagement of employees in the exercises, and the influence of teams on employee behaviour was observed and reported on by focus group participants:
“Many behaved in the same way...the whole team practised together.”

Many participants reported that the exercises were simple and easy to learn. The Qigong was valued as a type of exercise that could be easily undertaken in the workplace setting:
“We could practise the exercise at the workstations and not much space was required.”

Some liked the fact that the exercises were structured to target different parts of the body. The idea of incorporating Qigong seemed to appeal to most, but not all participants. One participant noted that it might have been useful to incorporate more movements from Tai Chi.

### 3.5. Maintenance

Maintenance is defined as the extent to which the intervention becomes part of routine organisational policy or practice, the long-term sustainability. Since long-term outcomes were not assessed in this study, we focus on perceptions of future sustainability, and mechanisms for future delivery. The majority of participants had shown appreciation of this health initiative. Employee enthusiasm for the Move-It intervention was evident in the animated discussion and the provision of ideas for ways to sustain the initiative in the longer-term. Members of the organisational committee indicated that the intervention could be sustained and integrated into long-term organisational policy:
“We should persist with the programme maintenance and reinforce employees’ interests in this initiative. Mind-set training is crucial.”

However, at the time of the focus groups senior management had not confirmed future investment in this health initiative. To sustain healthy behaviour beyond the intervention period, some participants suggested that new elements should be added in the pop-up windows every now and then, for example, by incorporating health facts or sharing employees’ experiences. Some suggested producing an integrated version of the exercise video, in addition to modular versions. Ideas about introducing competition, team-based approaches and incentives were also raised. Suggestions were offered about incorporating exercise into other company activities, such as sports days. The importance of building good habits was noted:
“We should take an exercise break regularly…develop it as a habit and be aware of its importance for health.”

## 4. Discussion

This is the first published study we are aware of that evaluates the outcomes and intervention implementation processes concerning a Qigong worksite exercise programme for sedentary workers in China. In this study, participants in both intervention and wait-list control groups reported significantly higher PA at T2 than at T1 (Objective 1), and the magnitude of this increase was similar in both groups.

There were no significant changes in work performance in the intervention group indicating no perceived negative impacts of the worksite exercise; a significant increase in self-reported work performance was found in the control group alone (Objective 2), and the increase in sitting time observed in both groups was significantly smaller in the intervention than the control group (Objective 1).

Process evaluation (Objective 3) revealed that the exercise intervention was successfully marketed to all employees, and implemented effectively with reasonably high uptake, and good adherence (as described by employees and managers) although the company managers reported that adherence declined over time. The high level of perceived organisational support and high acceptance of the intervention among employees suggested good adoption at both organisational and individual levels. There were some minor variations in the delivery of the intervention from that planned, although the intervention on the whole was fully delivered, in the correct order, and within the intervention timescale. Whilst there was enthusiasm for longer-term maintenance from employees and their managers, no concrete plan, for example in terms of resource allocation or administrative support was evident at the time of the process evaluation.

In workplace health programmes in practice, interventions tend to be accessible to all employees rather than offered to select groups with health risk factors. From an organisational perspective, employers commonly prefer to offer workplace health initiatives that are widely available to their workers, rather than highly targeted. Therefore, participants in this pragmatic wait-list control study were included by cluster (site) and were not randomised individually to groups in order to reflect common practice in workplace health promotion.

We found a positive change in PA in both groups, as has been shown in other wait-list studies with digital intervention for physical activity promotion [57]. The increase in PA in the control group may have been influenced by (a) a Hawthorne effect given the increased focus of the organisation on PA of all employees, associated with their participation in the research, or (b) the fact that preparatory work for the intervention for the wait-list control group started before the intervention, with team leaders being involved in the production of promotional videos, which may have led to a rise in PA levels in the control group before their intervention period started. Although this risk would have been minimised with an individually randomised controlled design, these designs are not always possible in real-world settings [58].

The importance of including worksite-related outcomes in evaluations of workplace physical activity interventions has been raised in a prior meta-analysis [59]. In our study, short bouts of Qigong exercise breaks did not improve self-reported work performance. Prior studies of workplace PA programmes have demonstrated inconsistent outcomes for organisational outcomes, and often focus on employee productivity rather than performance [37]; although we had an a priori rationale for measurement of performance, discussion with intervention sites indicated that objective productivity data would not have been available to us. We used a single item measure of work performance—the full HPQ may have been more responsive to change. Although, the timescale was relatively short for performance changes to take effect, and such changes are not easily observed in a population that has a relatively high baseline performance. Further, performance can be influenced by many other individual and organisational characteristics [60]. In future studies, the use of more proximal factors, such as job satisfaction or work engagement, might be indicative of the potentially helpful effects of such interventions at the organisational level [61]. Importantly, no differences were found in work performance with exposure to the intervention. This suggests that for employees taking time out to engage in worksite exercise twice during the working day, this did not impact negatively on their reported performance at work, e.g., the interruption of engaging in PA “did no harm”—it had no adverse effects on perceived work performance. This is important on two accounts: First, worker performance (and any associated changes in productivity) is an organisational priority; and second, employees reporting barriers to taking active work breaks have raised concerns about the potential for negative impact on their performance through interruption of work-flow [62].

At the study design stage, we intended to collect objective sickness absence data although these data were not accessible to the research team. However, there are contextual influences on the value of these data within our chosen setting since the organisation reported that sickness absence rates are always low, and China, in comparison to some countries, does not have high levels of reported sickness absence in its working population [63]. Additionally, the timescale of the intervention may not have been sufficient for any potential influence on absenteeism to become evident.

The process evaluation provided further context for these findings. The participation rate in this research from the total employee population was 40%, which is comparable with other similar studies. A systematic review suggested that participation levels in health promotion interventions at the workplace vary widely from 10% to 64% but are typically below 50% [64]. Those PA interventions with higher recruitment rates tend to incorporate interventions as part of the working day in paid time [65], as in the current study. Indeed, the actual participation rate may have been underestimated since the process evaluation revealed that workers who did not formally enrol may have taken part, either at work or at home.

This study used a participatory approach to the intervention delivery, by engaging team leaders as intervention facilitators; having an organisational committee of senior staff who were responsible for internal marketing and delivery of orientation and motivation briefing sessions for the team leaders; and many employees who volunteered assistance with technical support, production of materials and exercise demonstrations. As such there was high value attributed to the intervention by employees and managers alike. The quality of the intervention’s materials was valued, and there was good adoption of the intervention, by intervention group participants. Similar participatory practices have been used successfully in other studies to reduce sedentary time for office workers [66]. Regarding the nature of the intervention, while a few participants found the pop-up screens distracting, other participants expressed the need to be prompted by a computer or a person. Early research suggested that frequent prompts and personal contact generally enhance the effectiveness of health promotion initiatives [67], and more recently, employees exposed to a passive prompt were five times more likely to fully adhere to completing a movement break [68]. The use of prompts to motivate movement in sedentary office workers (via workstation screens, smartphone notifications or tactile feedback from wearable devices) has been discussed in a recent review around technology to reduce SB in office workers [28].

The attrition rate in the intervention group (27%) was fairly high, although a systematic review has demonstrated that a wide variability in attrition rates is found in digital workplace health research and raises challenges with sustaining engagement [29]. It is possible that the higher attrition rate in the intervention group may have influenced outcomes for PA, weekday sitting hours and work performance. Regarding intervention engagement, the most common reason for non-participation in the intervention was employee fear of over-commitment, and such fears have been interpreted in earlier studies as indicating both a lack of time and also low self-efficacy [69].

In terms of intervention fidelity, changes often happen in “real-world” settings [70] and may have unintended consequences [71]. The reasons why all team leaders did not all take a role in leading exercise breaks in this study, for example, is not fully understood. Other designated persons were able to adopt this role. Social support for PA from leaders or co-workers merits further investigation as a specific strategy to change employee behaviour [72]. The fact that the standard “reference” set of exercise demonstration videos were all posted at the start of the intervention instead of at intervals may have led to declined adherence to exercise because of a short span of exercise novelty; focus group participants did report that the novelty factor faded quickly.

Whilst enthusiasm was apparent, there was a lack of any concrete plan or clear message on programme maintenance at the organisation level. This contrasted with employees’ and employers’ expressions of interest in longer-term programme maintenance. Such findings are usually related to complexities in the organisational context [37]. For example, staffing changes affect organisational priorities and resource availability; both are commonly found to underlie differences between research and real-life commercial application [73]. Further exploration into employers’ motivations to maintain such initiatives is needed.

Although the intervention was delivered in 2013, these findings are of current value since increasing PA remains a public health priority, and workplace PA intervention is still not commonplace in Chinese workplace settings. Emerging evidence suggests that multi-component health promotion programmes might have health benefits (e.g., Taiwan [25]) although benefits for organisational outcomes such as work performance remain unclear and impacts on sickness absence are yet to be tested.

Due to the nature of the intervention it was not possible to blind employee participants or organisational managers to group allocation. To reduce potential for bias in collection of outcomes, data collection was done online and remotely, and individual-level outcome data were analysed by an independent researcher who was blinded. The study is limited by the small cluster size. The study was group randomised since individual-level randomised controlled trials are not always appropriate or feasible in the workplace setting [74]. The intervention itself was a pragmatic intervention that reflected real-world workplace interventions through accessibility to all employees with voluntary participation, rather than being offered to select groups with health risk factors. As such we have not assessed outcomes for groups of participants with particular health or behavioural characteristics and we are not able to determine whether our participants (mostly young, university graduates) were more motivated or more active than employees who chose not to take part, or employees in other types of organisation.

The focus group samples were small convenience samples, and the views expressed might not have reflected those of other employees and managers. As with all interview studies there may have been a risk of social desirability bias. It may be useful for future studies to consider collecting additional observational data relating to non-verbal communication and group interaction although this was beyond the scope of our study. Finally, focus group participants were drawn solely from the intervention site; input from the wait-list control site might have given additional insights, notably on why that group’s PA levels increased before delivery of the intervention.

The measures were self-reported and therefore subjective outcomes. As such it is not possible to determine objective changes in PA, sedentariness or work performance. However, IPAQ is a commonly used tool for assessment of PA, and some have argued that the use of pedometers or accelerometers, for example, may serve as interventions in themselves [75].

Despite scientific independence of the research team from the participating site there were some raw data that we were not able to access. We were not able to access objective data on work performance outcomes. We could not access the raw data for exercise adherence and as such we cannot compare “prescribed” exercise dose with actual dose and conduct per protocol analysis. However, other research (albeit in a culturally different setting) has shown that using per protocol analysis, physical training in the workplace can improve organisational outcomes [76]. Generally, researchers need to identify better ways of recording adherence and compliance in pragmatic workplace exercise intervention studies. With the lack of objective data, caution should be applied in the interpretation of outcomes. This trial highlights the challenges of collecting or accessing objective data in real-world organisations and this requires further investigation.

In conclusion, this study showed that delivery of a digital Qigong worksite exercise intervention was successful in raising awareness of the importance of PA and had wide reach and good uptake. Reported PA levels observably increased in both groups (perhaps reflecting the increase in promotion of PA across the organisation). The intervention had no perceived adverse effects on employee work performance through participants taking active work breaks. The intervention showed to be feasible and was acceptable to both managers and employees. The participatory approach was perceived positively, although the long-term commitment of the organisation to promoting exercise at work remains unknown. Despite its design limitations and the challenges of conducting worksite exercise interventions in real-world settings, this study has international relevance, and primarily contributes to a limited evidence-base on worksite exercise initiatives for the growing population of sedentary workers in China.

## Figures and Tables

**Figure 1 ijerph-16-03451-f001:**
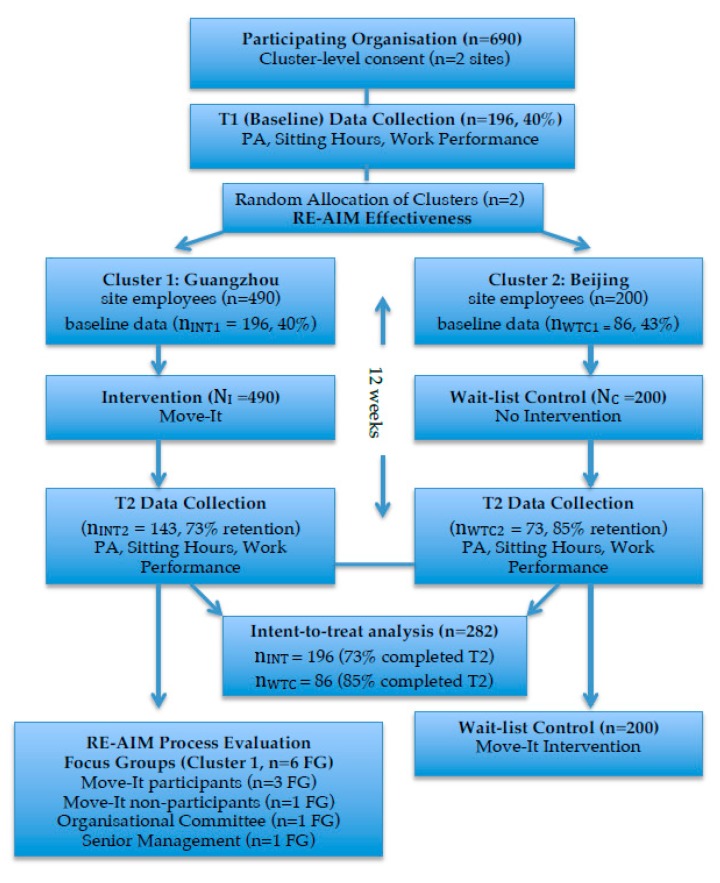
Study flow.

**Table 1 ijerph-16-03451-t001:** Link between components of Capability–Opportunity–Motivation-Behaviour (COM-B), Policy and Move-It intervention functions.

	Education	Persuasion	Training	Environmental Restructuring	Modelling	Enablement
**COM-B**						
C-Ph			X		X	
C-Ps	X	X	X			X
M-Re	X	X				
M-Au	X			X	X	X
O-Ph				X		
O-Soc		X		X		X
**Policy**						
Communication and Marketing	X	X		X		
Guidelines			X	X	X	
Regulation				X		X
Service Provision		X		X		

**Table 2 ijerph-16-03451-t002:** Participant characteristics by group.

Sample Characteristic (*n* = 282)	Intervention Group	Control Group
	*n* (%)	*n* (%)
All participants (baseline)	196 (69.5)	86 (30.5)
Gender		
Female	97 (49.5)	37 (43.0)
Male	96 (49.0)	46 (53.5)
Not specified	3 (1.5)	3 (3.5)
Education		
High school	5 (2.6)	0 (0)
College	20 (10.2)	1 (1.2)
University	168 (85.7)	82 (95.3)
Not specified	3 (1.5)	3 (3.5)
Marital status		
Single	58 (29.6)	19 (22.1)
Married	118 (60.2)	62 (72.1)
Neither single nor married	17 (8.7)	2 (2.3)
Not specified	3 (1.5)	3 (3.5)
Tenure		
<1 year	24 (12.2)	9 (10.5)
1–3 years	72 (36.7)	41 (47.7)
4–5 years	26 (13.3)	5 (5.8)
5–10 years	54 (27.6)	17 (19.8)
>10 years	17 (8.7)	11 (12.8)
Not specified	3 (1.5)	3 (3.5)

Note: Numbers are counts and percentages.

**Table 3 ijerph-16-03451-t003:** Pre- and post-intervention outcome variables (control and intervention groups).

	T1 (Means)	(SD)	T2 (Means)	(SD)	Changes (Intercepts^INT^)	(SE)	*p*	Difference I-C 95% CI (Lower; Upper)	SE	*d*	*p*
Physical activity (hours)											
Control	18.75	(19.75)	27.20	(30.84)	7.41	3.64	0.04	−1.61 (−9.76; 6.53)	4.16	0.05	0.70
Intervention	22.87	(21.84)	27.75	(25.98)	5.80	2.00	0.00				
Work performance^ALT^											
Control	7.63	(1.40)	7.50	(1.75)	0.69	0.24	0.01	−0.72 (−1.25; −0.19)	0.27	0.36	0.01
Intervention	6.89	(1.41)	6.96	(1.52)	−0.03	0.12	0.78				
Sitting hours											
Control	9.20	(2.20)	9.41	(2.01)	10.34	1.04	0.00	−4.66 (−7.25; −2.08)	1.32	0.44	<0.01
Intervention	9.51	(2.69)	9.64	(2.55)	5.68	0.82	0.00				

Note: Estimates come from: (i) a structural model with pre and post correlated for pre and post means; and (ii) latent change score (LCS) models with baseline predicting changes, for change estimates; SE = standard errors; d = Cohen’s d effect size; INT: the intercepts of the change scores are reported, estimated at the baseline control predictor average values; 95% confidence intervals are shown for the differences in changes; I = intervention, C = control groups; significant changes and differences bolded; physical activity is a total of walking, moderate and vigorous activity.

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
