# Peer review of "Move-It: A Cluster-Randomised Digital Worksite Exercise Intervention in China: Outcome and Process Evaluation"

_ijerph, 2019, doi:10.3390/ijerph16183451_

Round 1

Reviewer 1 Report

Thank you for the invitation to review this manuscript.

On the whole it is well written and while the intervention did not deliver as expected there are many lessons to be learnt from this study.

Below are some suggestions to improve the quality of the paper and information reported.

Introduction

Page 1, Lines 37 – 40: this is a very long sentence that doesn’t flow. It would read better if it was 2 sentences.

Page 2, line 3-4: This sentence needs a reference: “Sedentary behaviour (SB) is associated with adverse health outcomes independent of levels of physical activity (PA).”

Page 2, lines 1-3: the facts presented are interesting but the link between physical inactivity and hypertension and diabetes should be made clearer.

Page 2, lines 6-10. Again, a very long sentence that should be broken up. However, I question the relevance of the point being made as it does not link logically to the previous sentence.

Page 2, line 15: what does this sentence mean “Workplaces in China provide access to around three-quarters of the overall population”

Page 2, lines 29-32: this sentence is describing the findings of a review on the effectiveness of workplace interventions on SB and PA. However, all that is said is that “positive effects were found for PA/SB” . It would be helpful to the reader if the authors can expand on this without having to read the SR. Were the positive effects found for each of the 3 categories of interventions or was one category more effective than another?

Somewhere in the paper Qigong should be described as many people (like me) are unfamiliar with it? Is it like TaiChi?

Also, the authors need to indicate why this type of exercise was selected.

Page 3, lines 9 -14: The aims are explicitly stated but it is not clear anywhere in the paper if the outcomes of interest are overall physical activity or just physical activity at work.

Similarly for weekday sitting hours – will this include sitting hours at work and home/travel or just work hours?

Design

Could the authors please clarify why a wait-list control was selected for this study?

The duration of the intervention could be included here in the Design section.

Participants and procedures

The authors should provide more detail about the ‘large organisaton’ participating in this study. What industry sector did the organisation belong to – manufacturing, education, banking or health? Was the organisation a government or private company?

Please expand on 'nature of tasks'.

Page 3, line 26: add ‘years’ after giving the age range of participants.

Page 3 line 27: Although it is explained further in the paper, it would be useful to the reader to know in this section that a ‘cluster’ was a geographical location. Thus only 2 clusters.

In a way, it is good that all employees were eligible to participate, however, there does not appear to have been any screening undertaken. Is it possible that this exercise intervention would be contraindicated for some participants e.g. those pregnant or with arthritis or back pain?

How did the authors determine if all employees were safe to participate? If there are no contraindications to participation, it seems redundant to say that adverse events were recorded. How was an adverse event defined?

Page 4, lines 10-11: the sample size calculation is outlined but not which outcome it was based on? Was the sample size estimated to detect a difference of how much in which outcome? One would assume the primary outcome which in this was

Intervention

The exercises were to be performed at prescribed break times during the working day.

Could the authors provide more detail about the number of exercises performed in each session, number of repetitions for each exercise and number of times per day and number of days per week?

Apparently, video materials were developed with the team leaders at the participating sites. Did they select the exercise or just demonstrate them? If they selected the exercises, what criteria did they use to select them?

The Abstract actually gives the information needed ie employees performed “a 10-minute Qigong exercise session twice a day at set break times during the working day”. Please add this information to the manuscript.

Page 5, line 7-8: The six promotional videos were posted one every two weeks. Does this mean that employees did the one exercise for 2 weeks several times per day and then did the next exercise for 2 weeks? Were the exercises repeated in a certain sequence?

Page 5, line 31-32: information about the participants should be moved to the Participant section

Page 5, line 41: It looks like the Taiwanese version of the IPAQ was administered. Why not Chinese version? Could all employees read Taiwanese?

Page 5 – Measures.

There is no information provided about the baseline data collected. This information should be provided such as age, gender, body mass index, working hours, co-morbidities etc

Data Analysis

Page 6, line 17: the outcome variables were skewed and hence the data was transformed. Can the authors explain if the data was positively or negatively skewed?

There should be information about missing data – was it missing at random or not? How was missing data handled? Was multiple imputation performed?

Page 6 lines 19-21: I am not a statistician but I don’t think that t-tests are the appropriate analytical method to use to determine between group differences.

Also, t-tests do not allow for covariate adjustment for age, gender etc

Page 6, lines 30-32: There is no data comparing the baseline characteristics between the Intervention and Control groups.

Page 6 lines 40 – 42: A significant limitation of this study is that adherence data were collected by the participating sites and not made available to the researchers. The authors need to explain the nature of the relationship between the research team and organisation. Was this work undertaken as a paid contract service? Whose idea was it to conduct the research? Greater transparency in the funding of this research is needed to ensure scientific independence of the researchers.

Page 6, line 46-47: Apparently there were no adverse events reported. What was the reporting procedure for the adverse event? Was it to the line manager or research team?

Page 7, table 2 and 3: The data presented in these tables are the transformed values and thus are meaningless. I suggest showing the raw data but state the results are based on transformed data.

Process evaluation of this intervention: This section is well written.

Page 12, line 29: The authors suggest that a participatory approach was used to engage managers, supervisors and workers. I disagree. This study used a participatory approach to the "intervention delivery" only by engaging team leaders as intervention facilitators.

Participatory approach means that everyone is involved (workers, management) in the design and delivery of the intervention. In this study, only team leaders were involved in designing the delivery.

Page 12, line 44-45: The authors state that it was not possible to determine if those who dropped out differed to those who did not. This analysis can be done by comparing baseline demographics for those with data at the end of the study and those without.

Page 13, line 39: the word ‘determined’ does not need the ‘d’ at the end.

The Abstract succinctly and accurately reports the study and results. No changes needed.  

Author Response

Please see Reviewer 1 attachment.

Reviewer 2 Report

General comments

This is a pragmatic RCT conducted in a real life setting with all the difficulties and challenges this presents. The study contributes with important knowledge on implementation of a physical activity intervention among Chinese highly educated participants. The outcome of the study is negative but it is important that also knowledge provided by negative studies are disseminated.  In this study it may be specifically important to emphasize that physical exercise can be successfully implemented and that in spite of performing the physical activity in the paid worktime there was no negative influence on productivity.

The merits of the study is mostly in the systematic REAIM approach to the process analysis.  It is recommended to give this a much larger weight in the presentation leading to changes in the background as well as restructuring the method and result sections.  The effectiveness analysis is presented upfront but is really just a part of the REAIM.  It is recommended that the presentation of REAIM. currently given in the result section, should be moved to the method section. 

Introduction

The introduction aims at prevention on obesity, diabetes, high blood pressure etc.  But the intervention choosen, based on normalization theory and behavior change theory, focuses on short breaks with qigong with deep breathing and stretching in quite short daily periods for 12 months.  What is the program logic behind these choices?  How is it expected that qigong change sitting time, IPAQ and productivity?

I recommend emphasizing the strengths in this study, which may be the focus on implementation and measurements of productivity. An introduction could follow the logic that the non communicable diseases demand interventions, the workplace is an important setting but we need much more knowledge on implementation of different kinds of interventions for this specific setting.

Methods:

In the description of the intervention content it is hard to find the exact weekly time prescribed for the Qigong exercise as well as other details on the training.

Maybe the following recommendation on reporting of physical activity content could be of help:

Slade SC et al. Consensus on Exercise Reporting Template (CERT): A Modified Delphi Study. Phys Ther. 2016 Oct;96(10):1514-1524.

In the power calculation which of the outcomes was used for the calculation?  What is considered as the minimal important change? Was the use of only two large clusters considered in the power analysis? 

In the analysis section it is stated that only an intension to treat approach is taken. It would have been a real strength if the collected adherence data could have been used for a per protocol analysis. An important information from this trial is that we need better ways to record adherence and compliance in exercise intervention studies.  

Under the tables 2+3 it is stated that it is the transformed data that is presented. However, this is not clear for me compared to the description of the square-root procedure described in the analysis section?

Furthermore, it is not clear if this is a real “intention to treat analysis” as only those participating in both T1 baseline and T2 questionaire were analysed.  A “real” intention to treat would include all randomized participants and apply some kind of imputation for those not replying at T2. Most meta analysis only accept a difference between the two groups at T2 as an effect and does not even allow to adjust for differences at baseline since this is per definition random in a randomized trial.

Results

Since both groups increase in physical activity, this is probably due to other factors such as seasonal variations and this clearly illustrate the importance of a control group. So this should probably not be reported as a result in itself but rather a comparison between size of the change in the two groups should be presented?  In the current presentation of the results, there is no statistical comparison between the two groups. Even if there is no significant change from T1 to T2 within each group there could have been a significant difference between the changes of the two groups.

The reference to these results in the discussion should be modified.

Discussion

The discussion is relevant and insightful. It comments in a well reflected way on a number of the weaknesses in the study design such as self reported outcome, no real randomization, only two large clusters, contamination of the control group, non response bias, etc.

However, the major issue of a relevant individualized content of the intervention is not mentioned. In a health perspective and based on physiological evidence it should be questioned if the choice of qigong described as “a gentle exercise” the most effective content in the intervention?  Did the participants actually perform the exercise? If so, neither they or the company did get “value for money”  in terms of better health or more healthy lifestyle by investing time on Qigong.   

Considering the approach in the background one would have expected more emphasis on the lack of results that could improve or prevent lifestyle diseases.  

In the broad public discussion about promotion of physical activity to conquer non communicable diseases it is often assumed that all physical activity is effective and the more the better. But we do know from sports physiology that physical activity must be tailored to provide a specific effect.  In other words: what is the aim to have employees participate in a kind of physical activity that does not provide any benefit?.  And in the present study the beneficial effect is too small to be statistically significant, largest in the control group and no per protocol analysis was performed to evaluate if Qigong is effective.

In the conclusion the awareness of physical activity is really not measured in this study. The results of increased physical activity cannot be attributed to the intervention in accordance with your RCT design.

For a more comprehensive discussion about possibilities of RCT designs at workplace interventions the authors maybe could be inspired by the following two overview papers, one of them from a keynote on a Chinese Symposium on physical activity as medicine. Some of the studies in the reviews will corroborate your findings on productivity and maybe inspire further argues on why the workplace is so important a setting for health promotion.

Sjøgaard G, Christensen JR, Justesen JB, Murray M, Dalager T, Fredslund G, Søgaard K. (2016). Exercise is more than medicine: The working age population’s wellbeing and productivity. Journal of Sport and Health Science 5 (2016) 159–165 doi: org/10.1016/j.jshs.2016.04.004

Søgaard K and Sjøgaard G. (2017) Physical activity as cause and cure of muscular pain: evidence of underlying mechanisms. Exerc Sport Sci Rev. 2017 Jul;45(3):136-145.

In addition, the following study will be highly relevant to demonstrate that in a per protocol perspective physical training at the workplace can both increase productivity and decrease sickness absence.

 Just Bendix Justesen, Karen Søgaard, Tina Dalager, Jeanette Reffstrup Christensen, Gisela Sjøgaard. (2017). The Effect of Intelligent Physical Exercise Training on Sickness Presenteeism and Absenteeism Among Office Workers. J Occup Environ Med. 2017 Oct;59(10):942-948.

Specific comments:

Consider to rephrase the sentence: “Sedentary behaviour (SB) is associated with adverse health outcomes

independent of levels of physical activity (PA).”   In itself sedentary behavior is just the lowest level of physical activity. There is no reference given  but the sentence probably refers to some of the early studies showing that physical activity in leisure time did not compensate for occupational inactivity.  These studies did not use compositional data analysis that would have been the most correct analysis.  

The abbreviation PA for physical activity is not used consistently. Maybe just skip the abbreviation? And consider also SB that in many other context is Systolic Blood pressure.

In the design why not establish the intervention group and the waiting group at each of the two sites?

Why is the trial not   pre-registered on clinicaltrials.gov (ICT: To be inserted after peer review)?

It is unusual to comment on the representativeness in a qualitative study. It is a convenient sample and basically it is more about saturation of information. 

Reviewer 3 Report

A brief summary

The manuscript presents a cluster RCT aimed at increasing the physical activity (PA), reducing the sedentary time, and increasing the work performance of the participants. The intervention study was situated in China and consisted of employees who participated in an intervention offered by their employer at the worksite premises, in groups (departments), and during working hours. The intervention was delivered electronically using web and video in combination with live instructions of qigong activities and encouragement offered by team leaders. Both the intervention and the control group increased their self-reporting levels of PA but the differences were non-significant. There were no changes in sedentary time or self-reporting work performance. The implementation and reception of the intervention was evaluated by focus group interviews.

Broad and specific comments:

The majority of RCT on workplace health promotion are carried out in Europe and the USA. Sedentary lifestyle and related health complaints/diseases are global challenges. Intervention studies adapted to the local culture and working life in Asian countries are important both to the local health authorities and to the research field of health promotion worldwide. The study has a strong design being an RCT, and the cluster-randomization is suitable to the study aim and design of the intervention. The sample size and participation rate are both moderate, but the dropout rate is considerable and dose received in terms of attendance rate is unclear. The combination of protocol, intervention effect analyses and process evaluation is impressive and interesting, but ambitious. The information and analyses required to assess the quality and the replicability of the study is not complete, and some of the theoretical background is superficially presented. The authors should consider if the material should be separated into two publications.

Introduction and aim

The combination of quantitative analyses of intervention effects and qualitative analyses related to the process-evaluation is in my opinion unusual and interesting, but a bit ambitious. I find that important aspects of the theoretical and empirical background, the method, and the results are lacking. This is especially the case for work performance which is only mentioned briefly in the introduction without a proper definition, main findings (reviews and meta-analyses) related to effect of PA and PA interventions on work performance in general and specifically in China. I find that both in the Introduction and Discussion section, the terms work performance, productivity and workability is used interchangeably. Work performance is measured superficially with one item. A more detailed and sound explanation behind the choice of work performance as an outcome variable should be included. In the abstract, it is stated that the intervention is theoretically informed. It is not clear to me what part of the intervention you refer to here. Is it the digital delivery, the use of qigong, the groups-based aspect, the effects of the intervention on work performance. I suggest you consult the work of Michie and colleagues for guidance on how to state the theoretical, and not just the empirical, foundation of the intervention. For instance, Michie et al., 2016 I would urge the authors to consider whether the material presented in the present manuscript would benefit from being divided into two papers. This would allow for a more detailed and thorough presentation and increase the scientific value for the field of PA intervention studies. Alternatively, more of this information could be included in the present manuscript. In my opinion, this would require that other parts of the manuscript were rewritten to become more to the point. For instance, the study lacks data on the maintenance of the intervention, only the expressed intention to persist at the end of the intervention period. This section could be omitted. Page 3, lines 9-14: The hypotheses you tested should be stated clearly, not just the research questions.

Method

I commend the authors on the use and the inclusion of the CONSORT checklist. However, I find that a more detailed description of the power calculation procedure should be included in the methods section of the manuscript: which outcome variable did you include in the power calculations and did you define a level of change that you considered clinically relevant? What was the ICC set to and how did you estimate this value? Did you increase the sample size in order to compensate for possible dropout? Dropout is very common in intervention studies, even at workplaces. Consulting other similar studies of the same duration could help you estimate the dropout rate. Page 6, lines 1-5: The work performance is measures on a 10 point likert scale. In table 1, the score is 2.62 for the intervention group and 2.79 for the control group. If this is a mean score, the participants seem to rate themselves very low. The calculation of the score should be described in detail, and the result at baseline should be commented on. The dose of the intervention, according to protocol, should be stated clearly in terms of hours in total during the 12 weeks. If possible, also an estimate of the actual dose that those who participated on one or both assessments received. It is an interesting fact that several employees chose to participate on the activities but did not sign up as participants. But the main group to describe in terms of fidelity and attendance is those who signed up. Page 6, line 14-24: in an RCT you are interested to find out whether the between group change (condition) is stronger than the within group change (time, from T1 to T2). Using a paired sample t-test on each condition separately does not allow you to test the intervention effect relative to the control group. I recommend that the effects are analyzed using longitudinal analysis of covariance, multivariate and univariate analyses of variance repeated measures, or ANCOVA, Two-way ANOVA, MANOVA or mixed modelling. Page 6, line 14-24: I also recommend that an analysis of the effect sizes of the between group changes is included. Cohen’s d could be applied in this case. It is recommended and increasingly common that effect sizes are reported, and that the conclusions are not drawn solely on significance testing. Page 7, line 1: My impression is that you have not chosen not to analyze the possible differences between conditions at baseline on the demographic and outcome variables, nor are the possible differences controlled for in the analysis of intervention effects. This is a subject of a certain controversy, and knowledgeable statistician are currently debating for and against this practice. CONSORT does not state any preference here. I recommend Twisk, Bosman, (…) and Heymans (2018) paper which offers a fairly accessible argument pro this practice and even a test of different procedures for taking the significant differences into account in the analyses. I think you should consider doing a test of the baseline differences between conditions, especially because the randomization is weaker when it is on a cluster level with only two clusters, as you mention.

Results

In accordance with the CONSORT checklist, the manuscript includes an overview of the baseline assessments for each condition. The results should be commented on in the text as well in the Results section, like mean age, gender (%), their initial level of PA and sedentary time. I think that statistical analyses of the differences between the intervention and control group at baseline should be included and presented in table 1. The attendance rate (how many sessions did each participant attend during the intervention period) is unclear. These factors could possibly explain the lack of significant intervention effects. The dropout rate in the intervention group is substantial (around 27%). Given that fact that the analyses were done based on an intention-to-treat sample, I question how the missing data were handled in the analyses. Did you use any kind of imputation? And did you consider doing the analyses on a per-protocol or complete case sample (has both T1 and T2) as well? Since there are no significant between group changes, the effect of the intervention could possibly be weakened by a high dropout rate. I find the inclusion of the interview data a bit unusual, but it contributes to the discussion of the results. However, a thorough process evaluation should also include information about fidelity. In this case, including the mean number of times the participant actually took part in the exercises is important to consider the effectiveness of the intervention. You have chosen to use a one-item questionnaire to assess self-reported work performance. The HPQ contains several questions related to work and work performance in order to help the respondent make more accurate recollections and assessments of their own work performance. This is what the creators of HPQ call decomposition and the aim is to avoid superficial answers (Kessler et al., 2003). A comprehensive battery of questionnaires could explain the use of a single-item questionnaire on work performance, but this seems not to be the case in this study. I would like the manuscript to include more information about why these choices were made. The use of a single item questionnaire is commented on in the discussion section, and this is important. The use of a one-item questionnaire is discussed as a limitation, and this is a strength. But as already mentioned, the reasons behind this choice should be stated explicitly. In general, I think there should be more citations from the focus group interviews, especially in 3.2.3 Implementation.

Discussion

In general, I think the Discussion is good and a lot of the shortcoming of the study are reflected on. The additional analyses I have already commented on would strengthen the discussion with more facts. Page 11, lines 27-29: The results are compared to a South-African study. There are numerous studies and reviews of studies on workplace health and wellness promotion that are more similar to the present in terms of duration (12 weeks not 6), population (office workers not clothing manufacturing), the use of digital/video delivery (not pamphlets). I suggest that you either explain the similarities more explicitly or refer to studies that bear a stronger resemblance to the present. For instance related to the above mentioned characteristics, or related to countries that are closer to China culturally or studies using qigong. Page 12, lines 15-20: I like the argument and agree with the concussions which is in line with meta-analyses on the effects of workplace health promotion interventions which is mixed, see the meta-analysis of Conn et al., 2009. I suggest you use this reference in the manuscript.

Recommended papers:

Michie, S., Carey, R.N., Johnston, M., Rothman, A.J., de Bruin, M., Kelly, M.P. & Connell, L.E. (2016). From theory-inspired to theory-based interventions: A protocol for developing and testing a methodology for linking behaviour change techniques to theoretical mechanisms of action. Annals of Behavioral Medicine, 52(9), 501-512.

Twisk, J., Bosman, L., Hoekstra, T., Rijnhart, J. Welten, M. & Heymans, M. (2018). Different ways to estimate treatment effects in randomised controlled trials. Contemporary Clinical Trials Communications. 10, 80-85.

Round 2

Reviewer 2 Report

The paper has undergone a major revisions and has really improved.

The authors has answered all my concerns and done a good job in revision.

Author Response

Thank you for the time taken to review this manuscript.

We note that no further revisions are requested from this reviewer.

Reviewer 3 Report

Overall, the manuscript has been substantially improved and I commend the authors for including a statistician to reanalyze the data.

General comments:

I would use the word “participants” referring to the ones who completed one or two questionnaires, rather than “employees”. I would use “employees” or “eligible employees” for all those present at the worksites The study flow is much clearer I think the integration of process evaluation data and statistical analyses has improved the manuscript quality in terms of interpretation of the findings and the discussion.

Specific comments:

Pag 2, line 10-13: You state that the prevalence of sedentariness is increasing but you refer to a cross-sectional study. Could you specify the change for instance during the last 10 years? Page 3, lie 13: change the word “lifestyle PA” to “leisure-time PA” Page 3, line 15: omit the “to”, “to help promote” Page 3, line 16: would it be better to write “therapeutic effects of qigong”? Page 3, line: should be “with regard to general” Page 3, lines 19 – 21: I would expect an reference to studies on the effects of qigong or mindful approaches rather than the WHO. Wrong number? Page 3, line25: unclear, engagement in what? Work or qigong? Rewrite Page 4, lines 1-10: I think the argument of ethics should be included as well. It’s unethical to not offer the intervention to the control group as well. Page 4, lines 14-15: The sentence just repeats information from the former sentence and could be omitted. Page 4, section 2.2 (an in general): I would use the word “participants” referring to the ones who completed one or two questionnaires, rather than “employees”. I would use “employees” or “eligible employees” for all those present at the worksites Page 5, lines 8-12: You state that you expected an increase of 12 hours. Is that for the whole 12 weeks, one hour per week? The health recommendations regarding physical activity is normally 150 minutes of moderate-to-vigorous physical activity per week (see. https://www.who.int/dietphysicalactivity/factsheet_adults/en/). An increase of 60 minutes is a lot. You should refer to other studies or preferably meta-analyses to state a clinically relevant and realistic change. Page 5, lines 8-12: Why did you expect an increase in the control group when no intervention was offered during the period. I would expect zero change. Should be explained Page 7, lines 1-10: you give a detailed description of the qigong movements, but it is not clear to me whether they could be expected to increase PA intensity? Are the movements a mix of low and moderate intensity? Given that you use IPAQ to assess PA, intensity is an important aspect of the measurement of regular and health-promoting PA. Page 7, lines 2-4: The calculations are a bit unclear to me: 6 videos x 2 minutes = 12, not 10? Did they to through the 6 videos each day they were present at work? That would represent a dose of 12 x 5 workdays per week x 12 weeks = 720 hours. Were they expected to do the movements during weekend as well? Page 7, lines 15-17: How does this relate to the dose, see comment 15. Page 7, lines 22 – 49: I find that the systematic design of the intervention, using appropriate frameworks, is nicely described in detail. Page 8, line 8: Was the intervention delivered individually in terms of one-on-one, or in a group? Page 10, lines 1-22: I commend the authors on the comprehensive description the data analyses. I think that some of the information where you describe normality, skewedness and FIMl could have been shorter and more to the point as this is commonly used analyses. Page 10, lines 29-40: well written and important analyses to be included. I would include short information about significant differences between the intervention and the control group at baseline – not just between participants and eligible employees. Page 11, Table 2: It is common to state the descriptive data first followed by (n), and to indicate whether the data is mean, % etc. Page 11, line 26-28: having a control group, I think the main finding that should be reported first is the lack of intervention effect in terms of a significant between groups change. Secondly, the significant change from T1 to T2 should be reported. Page 12, table 3: I think effect size should be reported as well, there are online calculators available. In general, you report whether changes are significant or not, but a think you should comment on the clinically relevance of the changes as well – comparing them to a parameter such as health authority recommendations (PA) or a meta-analysis. Page 12, lines 5-7: long and unclear sentence, rewrite.
